# Prognostic value of FOXA1 in estrogen receptor-negative breast cancer: A systematic review and meta-analysis

Angela V. Fonseca-Benitez[1,2]*, David Díaz-Báez[3], James Guevara-Pulido[4], Milena Rondón-Lagos[5], Andrés Felipe Aristizábal-Pachón[1], Nelson Rangel[1]*

1 Departamento de Nutrición y Bioquímica, Facultad de Ciencias, Pontificia Universidad Javeriana, Bogotá, Colombia, 2 Universidad El Bosque, School of Dentistry, Bogotá, Colombia, 3 Universidad El Bosque, School of Dentistry, Unit of Basic Oral Investigation – UIBO, Bogotá, Colombia, 4 Universidad El Bosque, Facultad de Ciencias, INQA-Química Farmacéutica, Bogotá, Colombia, 5 School of Biological Sciences, Universidad Pedagógica y Tecnológica de Colombia, Tunja, Colombia

* fonseca_avictoria@javeriana.edu.co (AFVB); rangeljne@javeriana.edu.co (NR)

## Abstract

Breast cancer remains the most common cancer among women worldwide, and recurrence rates stay high despite current treatments, especially for those with negative estrogen receptor status, where therapies are less effective, and prognosis is worse. Identifying molecules with predictive value for therapy response and prognosis is therefore crucial. In this context, FOXA1 could serve as a potential biomarker to predict the progression of ER-negative tumors. A search was conducted to answer the question, "What is the prognostic value of FOXA1 expression in breast cancer, estrogen receptor negative?" using various databases. Controlled vocabulary and Boolean operators were employed. Only studies reporting overall survival and disease-free survival, defined as the time from evaluation to death or relapse, were included. We identified seven articles evaluating FOXA1 and its relationship with disease-free survival (DFS) or overall survival (OS) in patients with ER-negative breast cancer. Our data indicate that higher FOXA1 expression is associated with improved overall survival (HR = 0.61, CI = 0.45–0.83, p < 0.002) and better disease-free survival (HR = 0.69, CI = 0.51–0.93, p < 0.02). These findings suggest that FOXA1 is linked to a favorable prognosis in terms of overall survival and disease-free survival. Further studies are needed to assess the role of FOXA1 in response to chemotherapy. PROSPERO registration number: CRD42024453750

### Key messages:

### What is already known about this topic?

Some studies suggest a link between higher FOXA1 levels and better prognosis in Luminal ER-positive Breast Cancer. Although FOXA1's role in hormone-independent environments is not fully understood, few studies indicate

**Data availability statement:** All relevant data are within the manuscript and its Supporting Information files.

**Funding:** This work was supported by the Foundation for the Promotion of Research and Technology - Banco de la República (Grant No. 5,038) and the Ministerio de Ciencia, Tecnología e Innovación de Colombia (National Doctorate scholarship for Teachers, Call 909-2022). The funders had no role in study design, data collection and analysis, decision to publish, or preparation of the manuscript.

**Competing interests:** The authors have declared that no competing interests exist.

it likely plays a significant role in carcinogenesis due to its function as a pioneer factor.

## What this study contributes

Our results showed that higher levels of FOXA1 are associated with increased overall survival and disease-free survival, therefore, indicating a better prognosis in estrogen receptor-negative cancers patients.

## How this study could impact research, practice, or policy

Our findings emphasize the potential of FOXA1 as a prognostic marker in ER-negative patients. Based on our results, we recommend further research on this topic, especially focusing on therapy response in different Breast Cancer subtypes, particularly hormone-independent types, which often have worse survival outcomes. These findings underscore the importance of exploring new treatment strategies that target FOXA1 and other emerging targets.

## Introduction

Breast cancer is one of the most common cancers worldwide. It continues to have the highest incidence among women for all cancers [1]. Currently, the resistance rate to therapy in breast cancer tumors is increasing, with some cases being much less susceptible to chemotherapeutic agents than others [2]. Specifically, estrogen receptor-negative subtypes that do not express progesterone receptor (PgR) and human epidermal growth factor receptor 2 (HER2) have a higher recurrence rate, a lower response to chemotherapy, and are considered to have a worse prognosis. This can be explained because breast cancer is a heterogeneous disease with variable outcomes, making current biomarkers insufficient to predict therapy response and survival in breast cancer patients. Therefore, it is crucial and timely to discover new biomarkers for predicting breast cancer progression. One such biomarker is FOXA1, a pioneer factor and the founding member of the forkhead family. This family includes 40 highly conserved genes involved in various cellular processes, especially those influenced by hormones [3]. In this context, the pioneering activity of FOXA1 has been linked to chromatin remodeling, which facilitates access to and interaction with the promoter regions of several targets, including nuclear receptors of high clinical relevance, such as the androgen receptor (AR) and estrogen receptor [4]. FOXA1 has been extensively studied in ER-positive breast cancer, showing a correlation between its positive protein expression (FOXA1+) and favorable clinical characteristics such as smaller tumor size and lower grades.

Consequently, most studies suggest a link between elevated FOXA1 levels and better prognosis in Luminal (ER+) [5–7]. Although research on this topic is limited, even within ER-positive breast cancer, reports indicate that FOXA1 likely plays an important role in carcinogenesis, due to its function as a pioneer factor [8]. However,

its effect on prognosis in hormone receptor-negative environments (ER-negative) has not yet been explored. Therefore, this study conducted a systematic literature review (RSL) to examine the relationship between FOXA1 expression and breast cancer prognosis in ER-negative patients. The main question of this study was: What is the prognostic value of FOXA1 expression in ER-negative cases?

## Methods

### Protocol and registration

The current systematic review was designed following the PRISMA (Preferred Reporting Items for Systematic Reviews and Meta-Analyses) guidelines for systematic reviews. Additionally, it was registered in the International Prospective Register of Systematic Reviews (PROSPERO) (http://www.crd.york.ac.uk/prospero/) with the registration number CRD42024453750.

### PECO question structure and search strategy

The RSL was based on a question structured using the PECO framework: P (Problem): Breast cancer, E (Exposure): FOXA1, C (Comparison): Absence of FOXA1 protein or gene expression, and O (Outcome): Survival or prognosis. The question addressed in this study was: What is the prognostic value of FOXA1 expression in ER-negative cases? Extensive research was conducted using PubMed, Embase, Scopus, Google Scholar, BVS, and Cochrane to answer the PECO question. Controlled vocabulary included MeSH, Emtree, and relevant accessible terms, combined with Boolean operators "AND" and "OR." No limits were set for date, publication type, or language. Additionally, manual searches were performed using the snowball method, the PubMed "similar" tool, and www.connectedpapers.com. The search strategies are detailed in S1 Table. The searches were carried out in July 2025.

### Selection criteria

Studies meeting the following criteria were considered eligible for this RSL: first, observational case-control or cohort clinical studies. Second, studies whose results show the gene and/or protein expression of FOXA1 assessed through polymerase chain reaction (PCR) or immunohistochemistry (IHC), respectively. Regarding the results, studies reporting at least one of the primary outcomes were included, including those measuring Overall Survival in terms of relapse time from the date of evaluation to death, regardless of cause, and studies reporting disease-free survival. As secondary outcomes, studies on disease recurrence after treatment with different chemotherapeutic agents were also evaluated.

### Screening and selection studies

For screening and selection studies, Mendeley 19.4.8 was used to analyze and remove duplicates from search results from different sources. Then, the titles and abstracts were independently and blindly reviewed by two reviewers (VF-B & JG-P) using the web version of Rayyan software https://www.rayyan.ai/ [9] to identify potentially relevant studies that met the previously described eligibility criteria. Any disagreements between reviewers were resolved through consensus or with the help of a third evaluator (NR). Finally, full-text documents of all potentially relevant citations were obtained, and their compliance with eligibility criteria was again independently and blindly validated. Any discrepancies in the final selection were resolved by consensus.

### Quality assessment of the included studies

The article quality was assessed independently and blindly using the Newcastle Ottawa Scale (NOS) assessment tool [10]. The quality was categorized as follows: low quality (−4), moderate quality (5–6), and high quality (>7).

## Data extraction

Risk ratios and their 95% CIs were used to measure the association between FOXA1 and the survival of ER-negative patients. The data extraction was performed independently and blindly, following a predetermined Excel template that included the following information: author, year, type of study, type of breast cancer, population age, FOXA1 measurement method, primary outcomes, and secondary outcomes.

## Synthesis and statistical analysis of data

Initially, a narrative synthesis of the information was conducted using the finding tables. The effect of FOXA1 (gene or protein) presence on disease-free survival and overall survival of ER-negative patients was assessed using the Hazard Ratio (HR) and the 95% confidence interval (CI). A general random-effects inverse variance meta-analysis was performed to estimate the pooled prognostic risk. Statistical heterogeneity between studies was assessed with the I2 (square) test, where values greater than 30 (I2 > 30) indicated high heterogeneity. All analyses were conducted at a 95% confidence level using RevMan 5.4.1 software (Cochrane Training site based in London, UK). Due to limited study availability, a sub-group analysis was not performed.

## Ethical approval

This article does not include any studies with human participants or animals conducted by the authors and only presents the results of other researchers.

# Results

## Search and study the selection process

Using the search strategy described above, a total of 788 potentially eligible articles were identified from the various databases used: Embase (402), PubMed (102), Scopus (106), BSV (130), Google Scholar (81), Lilacs (0), and Cochrane (20). 161 articles were removed during the duplicate review process. 627 documents were screened by title and abstract for evaluation, with 531 excluded for not meeting the eligibility criteria. Eventually, 96 articles were evaluated in full text for final review, with 8 studies included [11–18] (Fig 1). However, one of these was excluded because the data required for the review was not accessible. The studies excluded during the final selection are listed in S2 Table.

## Description of the risk of bias assessment for the included studies

The risk of bias assessment of the included studies showed that five articles had a low risk, based on the referenced quality assessment scale. Regarding patient selection, the analysis revealed that six out of seven studies had a high risk in the item involving the selection of the non-exposed cohort. However, five out of seven articles demonstrated low risk concerning comparability. Similarly, the outcome evaluation showed that six out of seven studies indicated low risk, and all studies had adequate cohort follow-up due to the timing of patient assessments (Fig 2).

## Characterization of the studies included

The included studies were published between 2009 and 2023; four were from China, one from Italy, one from Portugal, and one from Japan. They mainly consisted of cohort studies, with four being prospective and three retrospectives. In total, 2436 patients aged between 40 and 80 years were included, with follow-up times ranging from a minimum of 5 years to a maximum of 17 years. Significant variability was observed in the assessment of histological grade, lymph node status, Ki-67 marker status, and presence or absence of metastasis. Regarding tumor size, most tumors were >2 cm. Similarly, negative lymph node status was the most frequently reported parameter. Only two studies considered the treatment received by patients, but these results were not linked to FOXA1 expression. The methods used to evaluate FOXA1

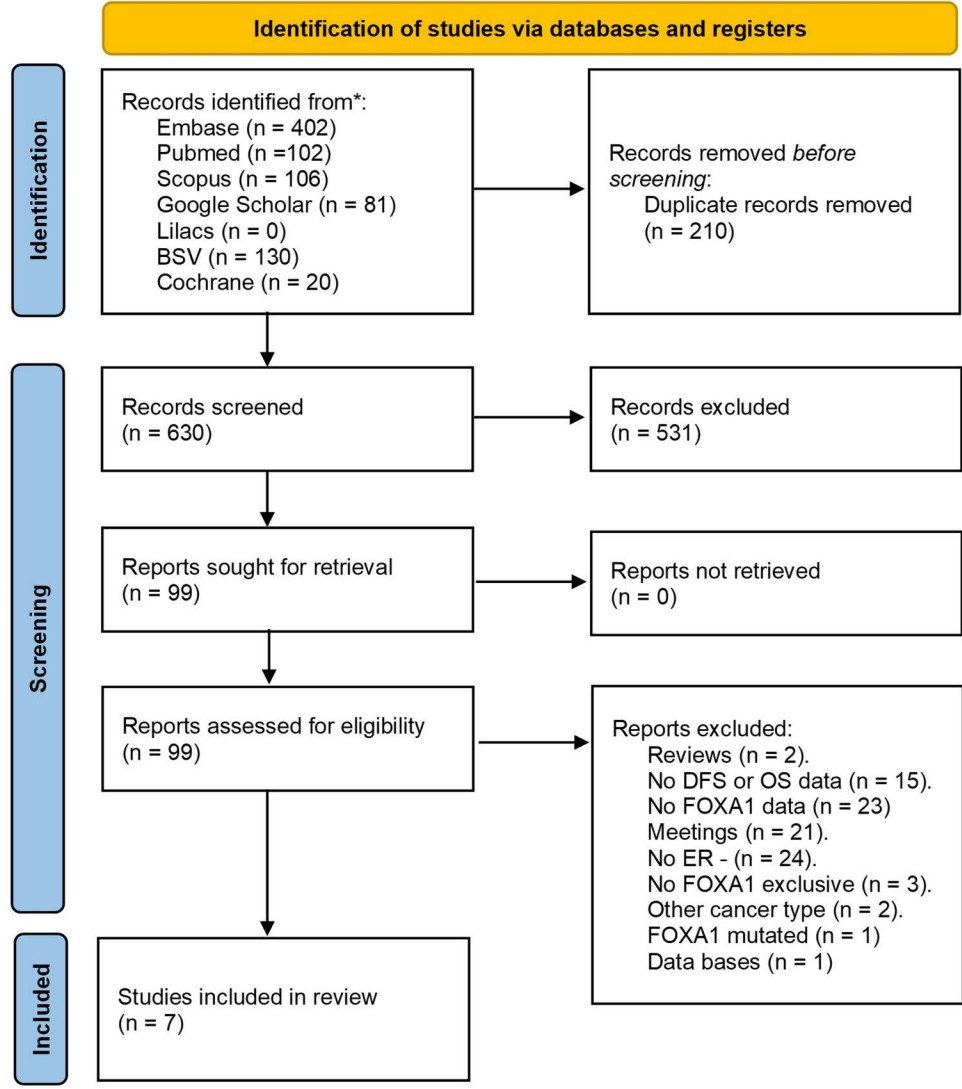

**Fig 1. PRISMA flow diagram for study selection.**

included both immunohistochemistry (IHC) and PCR. Nevertheless, protein expression analysis was used in most studies, while PCR was performed in only one study (Tables 1 and 2).

### Relationship between FOXA1 and ER-negative prognosis

Three (Albergaria, 2009; Dai, 2019; Chen, 2021) of the seven articles (543 patients evaluated) described overall survival values by assessing FOXA1 using IHC. The cut points for each study are shown in Fig 3. Three out of four independent studies demonstrated a clear trend (significant association) toward longer overall survival in ER-negative patients who tested positive for FOXA1 protein expression (Chen, 2021; Dai, 2019; Albergaria, 2009). Furthermore, a meta-analysis of all three studies showed that ER-negative patients who tested positive for FOXA1 expression also had increased overall survival (HR = 0.61, CI = 0.45–0.83, p = 0.002).

**Fig 2. Bias assessment.** The heat map related to the risk assessment is displayed. It compares the items evaluated for each study and their quality. Newcastle-Ottawa Scale (NOS): low risk (Green) and high risk (Red).

In addition, seven of the included articles reported Hazart Ratio (HR) disease-free survival data (Fig 4). When analyzing the documents, regardless of the method of FOXA1 assessment (protein or mRNA), it was noted that only two studies indicated a clear trend (significant association) toward extended disease-free survival in ER-negative patients who were positive for FOXA1 expression. Only one study (Mangia, 2019) reported that FOXA1 gene expression was related to a worse prognosis. Despite the above, upon meta-analysis of six studies (Chen, 2021; Dai, 2019; Hisamatsu, 2015; Mangia, 2019; Jing, 2019; Zhou, 2023), it was observed that ER-negative patients who tested positive for FOXA1 expression were also associated with increased disease-free survival (HR = 0.69, CI = 0.51–0.93, p < 0.02). To assess potential bias introduced by the fact that only one study (Mangia et al.) accounted for menopausal status, we performed a sensitivity analysis excluding this study. The results remained consistent with the primary meta-analysis (HR = 0.65, 95% CI: 0.48–0.87, p = 0.0052), supporting the robustness of our overall findings (S1 Fig).

## Discussion

This review, which is, to the best of our knowledge, the first of its kind, aims to analyze the relationship between FOXA1 protein expression and the prognosis of ER-negative patients. Our groundbreaking results show that FOXA1-positive levels are associated with better overall survival and disease-free survival outcomes. Regarding the existing literature, studies that evaluate FOXA1 expression through PCR or IHC in ER-negative tumors are very rare, with only seven articles published up to July 2025. However, one of these lacks complete data. IHC was the main method used in all the studies reviewed. In terms of study quality, most (5/7) had a low risk of bias. The areas where studies generally scored lower involved cohort selection not exposed and, to a lesser extent, the comparability of the cohorts evaluated in each case.

Table 1. Summary of demographic data for all included studies.

| Author, Year | Country | Study type | # cases RE (-) | Age | Time (Years) | Histological grade | Tumor size | Lymph nodal stage | Ki-67 | Menopause status | Metastasis | Clinical stage | Treatment | Method | PR status | HER2 status | Outcome |
|---|---|---|---|---|---|---|---|---|---|---|---|---|---|---|---|---|---|
| Albergaria, 2009 | Portugal | Prospective cohort | 107 | 57±14.2 | 10 | Grade I 51 (20.5) Grade II 116 (46.6) Grade III 82 (32.9) | 32±21 mm | Negative 111 (44.6) 1 to 3 lymph nodes 57 (22.9) >3 lymph nodes 54 (21.7) Not assessed 27 (10.8) | NI | NI | NI | NI | Surgery | IHC | Positive 89 (35.8) Negative 154 (61.8) Unknown 6 (2.4) | Positive 42 (16.9) Negative 201 (80.7) Unknown 6 (2.4) | OS |
| Hisamatsu, 2015 | Japan | Prospective cohort | 49 | 55.0±1.2 | 7 | NI | ≤2 Low-FOXA1 42 (39.3) HighFOXA1 61 (57.0) 2</≤5 LowFOXA1 53 (49.5) HighFOXA1 36 (33.6) 5<Low-FOXA1 12 (11.2) HighFOXA1 10 (9.4) | NI | LowFOXA1 0.20±0.01 HighFOXA1 0.14±0.01 | NI | Negative Low 56 (52.3) High 65 (60.7) Positive Low 51 (47.7) High 42 (39.3) | Stage I, 116 (54.2%) Stage II, 116 (54.2%) Stage III, 27 (12.6%) | Aromatase Inhibitors | IHC | Negative Low 59 (55.1) High 35 (32.7) Positive Low 48 (44.9) High 72 (67.3) | Negative Low 78 (72.9) High 90 (84.1) Positive Low 29 (27.1) High 17 (15.9) | DFS/OS |
| Dai, 2019 | China | Prospective cohort | 82 | <56 41 ≥56 41 | 5 | I 3 II 41 III 38 | ≤2 31 ≥2 51 | NI | NI | NI | NI | I+II 37 III+IV 45 | NI | IHC/qPCR | TNBC | TNBC | DFS |
| Mangia, 2019 | Italia | Retrospective cohort | 124 | ≤51 years 65 (52.4) >51 years 59 (47.6) | 17 | G1 1 (0.8) G2 23 (18.7) G3 99 (80.5) Unknown 1 | ≤2cm 59 (48.0) >2cm 64 (52.0) Unknown 1 | Negative 65 (54.2) Positive 55 (45.8) Unknown 4 | Negative (≤20%) 12 (9.8) Positive (>20%) 110 (90.2) Unknown 2 | Pre 83 (66.9) Post 41 (33.1) | NI | NI | NI | IHC | TNBC | TNBC | DFS/OS |
| Jing, 2019 | China | Retrospective cohort | 1583 | ≤51 1492. >51 2263 | 10 | NI | NI | Negative 2447 Positive 1761 | NI | NI | NI | NI | NI | PCR | Negative – 1076 Positive – 1545 | Negative – 1596 Positive – 217 | DFS |

*(Continued)*

**Table 1.** (Continued)

| Author, Year | Country | Study type | # cases RE (-) | Age | Time (Years) | Histological grade | Tumor size | Lymph nodal stage | Ki-67 | Meno-pause status | Metasta-sis | Clinical stage | Treat-ment | Method | PR status | HER2 status | Out-come |
|---|---|---|---|---|---|---|---|---|---|---|---|---|---|---|---|---|---|
| Chen, 2021 | China | Pro-spective cohort | 263 | ≤40 247 (23.7) 41–60 674 (64.7) ≥61 120 (11.5) | 8 | I/II 695 (73.3) III 253 (26.7) Unkonwn 93 | <2 296 (29.7) ≥2 702 (70.3) | Yes 561 (56.2) No 437 (43.8) Unknown 43 | ≤14% 247 (25.2) >14% 735 (74.8) Unknown 59 | Pre 585 (59.2) Post 403 (40.8) Unknown 53 | Yes 28 No 966 Unknown 47 | I 174 (17.5) II 517 (52.1) III/IV 302 (30.4) Unknown 48 | NI | IHC | Negative 269 (27.2) Positive 721 (72.8) Unknown 51 | Negative 547 (61.3) Positive/equivocal 345 (38.7) | OS |
| Zhou, 2023 | China | Retro-spective Cohort | 228 | 48 ±42–56 | 8 | I/II 620 (73.8) III 220 (26.2) Missing 75 | ≤ 2 282 (30.8) > 2 633 (69.2) | Negative 421 (46.0) Positive 494 (54.0) | NI | NI | NI | I 169 (18.5) II 493 (53.9) III 253 (27.7) | NI | IHC | Negative 237 (27.0) Positive 641 (73.0) Missing 37 | Negative 614 (66.5) Equivocal 77 (8.2) Positive 224 (25.3) | DFS |

*NI: No information

**Table 2. Summary of FOXA1 expression and DFS results from included studies.**

| n° | Author, Year | Cut Point | FOXA1 | OS (HR) | LONG RANK (p-Value) | DFS (HR) | LONG RANK (p-Value) | High-lights | Bias of Risk |
|---|---|---|---|---|---|---|---|---|---|
| 1 | Albergaria, 2009 | 0 = Negative<br>1 = 1–10% positive<br>2 = 11–20% Positive<br>10 = 91–100% Positive | Positive<br>Negative | 1<br>3.61 (0.83 to 15.60) | 0.086 | NI | NI | Better | low |
| 2 | Hisamatsu, 2015 | 71.70% | Low (<71.7%)<br>High (>71.7%) | NI | P = 0.0002 | 1<br>2.77 (1.26–6.37) | 103 | Better | Unclear |
| 3 | Dai, 2019 | Score 0 0–3 < 5<br>Score 1 1 < 80<br>Score 1 2 < 40<br>Score 2 1 ≥ 80<br>Score 2 3 < 40<br>Score 3 2 ≥ 80<br>Score 3 3 ≥ 4<br>High levels of FOXA1 (score≥2). | mRNA | 0.66 (0.58–0.74) | 1.70E-11 | 1<br>0.88 (0.68–1.13) | 0.320 | Better | low |
| 4 | Mangia, 2019 | Negative <10<br>Positive ≥10 | Positive<br>Negative | NI | NI | 0.63<br>(0.27 −1.47)<br>1 | 0.281 | Bad | low |
| 5 | Jing, 2019 | NI | Low<br>High | NI | NI | 1<br>1.08 (0.87-1.34) | 0.500 | Better | Low |
| 6 | Chen, 2021 | 0 y 300 (Continuos variable)<br>H score | ≤270<br>271-285<br>≥285 | 1<br>3.00 (1.25 - 7.22)<br>1.25 (0.65 - 2.39) | 0.19 | 1<br>2.46 (1.38 −4.37)<br>1.52 (0.96 −2.42) | 0.880 | Better | Low |
| 7 | Zhou, 2023 | 0 y 300 (Continuos variable)<br>H score | Low<br>High | NI | NI | 1.00 (reference)<br>2.60 (1.15 - 5.88) | 0.526 | Better | Low |

*NI: No information

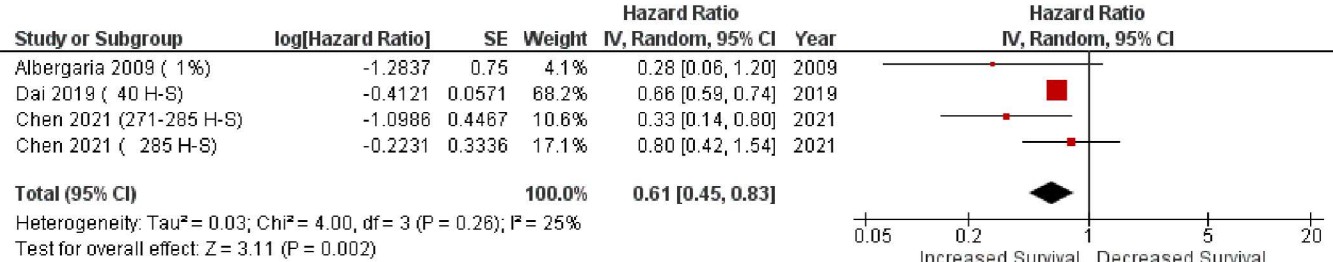

**Fig 3. Forest plot of overall survival (OS) based on FOXA1 expression in estrogen receptor-negative breast cancer.** CI: Confidence interval. The cutoff points for quantifying FOXA1 are provided along with each author.

During breast development, FOXA1 is expressed in a coordinated manner with Estrogen-Receptor, highlighting the importance of FOXA1 in the gland's morphogenesis and, consequently, in breast cancer tumorigenesis. Studies indicate that its gene expression is essential for maintaining luminal phenotypes. This underscores the clinical relevance of FOXA1, especially since Estrogen-Receptor expression has been linked to a better prognosis and endocrine response [19–20]. Additionally, recent research emphasizes FOXA1's role in gene regulation during both physiological and tumor processes [21,22]. Similarly, in ER-negative cases, variations in FOXA1 expression have been associated with the

| Study or Subgroup | log[Hazard Ratio] | SE | Weight | Hazard Ratio IV, Random, 95% CI |
|---|---|---|---|---|
| Chen 2021 (271-285 H-S) | -0.9002 | 0.2949 | 13.1% | 0.41 [0.23, 0.72] |
| Chen 2021 ( 285 H-S) | -0.4187 | 0.2345 | 16.0% | 0.66 [0.42, 1.04] |
| Dai 2019 ( 40 H-S) | -0.1222 | 0.1276 | 21.8% | 0.88 [0.69, 1.14] |
| Hisamatsu 2015 ( 71.7%) | -1.0188 | 0.4019 | 9.2% | 0.36 [0.16, 0.79] |
| Jing 2019 (High) | -0.077 | 0.1103 | 22.7% | 0.93 [0.75, 1.15] |
| Miangia 2019 ( 10 H-S) | 0.462 | 0.4323 | 8.4% | 1.59 [0.68, 3.70] |
| Zhou 2023 (High) | -0.9555 | 0.4155 | 8.8% | 0.38 [0.17, 0.87] |
| **Total (95% CI)** | | | **100.0%** | **0.69 [0.51, 0.93]** |

Heterogeneity: Tau² = 0.09; Chi² = 17.88, df = 6 (P = 0.007); I² = 66%
Test for overall effect: Z = 2.42 (P = 0.02)

**Fig 4. Forest plot of Disease-free survival (DFS) based on FOXA1 expression in estrogen receptor-negative breast cancer, with CI indicating confidence interval.** The cutoff points used to quantify FOXA1 are provided along with each author.

activation or suppression of different signaling pathways, suggesting a significant role for FOXA1 in this breast cancer subtype.

Our review showed that FOXA1 presence was not dependent on ER expression [23]. The above is because FOXA1 expression was consistently found in ER-negative and ER-positive tumors. To emphasize this, FOXA1 expression also seems to be related to increased survival in both ER-negative and ER-positive tumors. This agrees with studies that have determined FOXA1 influences the phenotypic plasticity of basal and luminal breast cancer cells by repressing basal markers, regardless of the presence of estrogen receptors. Furthermore, it has been reported that FOXA1 plays a regulatory role in apoptotic processes through AGR2 in other types of cancer, suggesting that FOXA1 could be a potential therapeutic target. However, it is unclear whether inhibiting it could promote tumor transformation [24–26].

Regarding treatment, we did not find studies that evaluated the prognostic value of FOXA1 in response to any chemotherapeutic agent in ER-negative patients. However, previous studies in ER + /HER2- luminal breast cancers showed that decreased FOXA1 expression predicted a better response to neoadjuvant treatment, thus recommending neoadjuvant chemotherapy for patients with low FOXA1 expression [27]. In this context, the outcomes of these studies lack significance regarding treatment response, as earlier reports indicate that FOXA1 overexpression can forecast resistance to therapy, including immunotherapy and chemotherapy, not only in ER-positive breast cancer but also in prostate and bladder cancer. This could also be explained by the high heterogeneity of the disease across multiple subtypes, which results from the variety of expression profiles of diverse receptors that can influence treatment response [28].

It is important to note that, based on the available information, no studies have reported a link between FOXA1 expression and treatment response in ER-negative cases. Our meta-analysis still has some limitations related to the available literature. First, the data reported in the included studies was highly heterogeneous. Additionally, different cut-off values were used to determine FOXA1 expression. Most studies detected FOXA1 expression using Immunohistochemistry (IHC). However, variations in antibody types, concentrations, and the subjective nature of interpreting IHC results could contribute to the heterogeneity observed across these studies. Despite these issues, our results provide an approximation of FOXA1's role in ER-negative breast cancer. This is the first review systematically analyzing the association between FOXA1 and breast cancer in hormone-independent environments. Our findings showed that higher FOXA1 expression in ER-negative patients was associated with a better prognosis in terms of overall survival and disease-free survival.

## Limitations

Systematic reviews are inherently limited by the quality of the primary studies included. A major limitation is the high heterogeneity among current studies, especially in the techniques and protocols used for FOXA1 determination (such as methodologies, antibodies, and cutoff values). Additionally, there are significant differences in the statistical analyses performed in these studies, making comparison and pooled interpretation of results challenging. Another important limitation is that the impact of FOXA1 expression on posttreatment prognosis in hormone-independent settings is not yet fully understood. Therefore, we recommend future standardized studies to address these gaps. These studies should focus on assessing chemotherapy response across different breast cancer subtypes, particularly aggressive phenotypes such as basal-like or triple-negative cancers, which have lower survival rates. Ultimately, larger, well-defined cohorts are needed to strengthen the validity of these findings. As a result, the conclusions of this study should be viewed in light of its limitations. The applicability of the results is restricted by variability in FOXA1 assay methodologies and scoring criteria, so these findings should be interpreted with caution.

## General recommendations

Research based on omics data is needed that doesn't restrict current publications and enables a more comprehensive understanding of relevant biomarkers, such as FOXA1, in complex diseases like estrogen receptor-negative breast cancer. Additionally, larger cohort studies are necessary to explore the relationship between FOXA1 expression and breast cancer prognosis in hormone-independent settings, as well as its connection to treatment response, using standardized assessment methods.

## Supporting information

**S1 Table. Database Search Term.**
(PDF)

**S2 Table. List of excluded items and reason for exclusion.**
(PDF)

**S3 Table. PRISMA CHECKLIST 2020.**
(PDF)

**S1 Fig. Sensitivity analysis.** Forest plot of Disease-free survival as a function of FOXA1 expression in estrogen receptor-negative breast cancer CI: Confidence interval. Excluded.
(PDF)

## Author contributions

**Conceptualization:** Angela V. Fonseca-Benitez, Andrés Felipe Aristizábal, Nelson Rangel.

**Data curation:** Angela V. Fonseca-Benitez, David Díaz-Báez, James Guevara-Pulido, Milena Rondón-Lagos.

**Formal analysis:** Angela V. Fonseca-Benitez.

**Methodology:** Angela V. Fonseca-Benitez, David Díaz-Báez, James Guevara-Pulido, Nelson Rangel.

**Writing – original draft:** Angela V. Fonseca-Benitez.

**Writing – review & editing:** Milena Rondón-Lagos, Andrés Felipe Aristizábal-Pachón, Nelson Rangel.

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
