## [Decision Letter · Decision Letter 0]

19 Aug 2025

Dear Dr. Rangel,

We look forward to receiving your revised manuscript.

Kind regards,

Yan Xu, Ph.D.

Academic Editor

PLOS ONE

Journal Requirements:

Fundación para la Promoción de la Investigación y la Tecnología - Banco de la Republica Nº 5.038 and by Pontificia Universidad Javeriana through the “Support for the publication of high-quality research articles 2024” grant

4. Please update your submission to use the PLOS LaTeX template. The template and more information on our requirements for LaTeX submissions can be found at http://journals.plos.org/plosone/s/latex.

6. Please include your tables as part of your main manuscript and remove the individual files. Please note that supplementary tables (should remain/ be uploaded) as separate "supporting information" files.

7. Please remove all personal information, ensure that the data shared are in accordance with participant consent, and re-upload a fully anonymized data set.

Reviewers' comments:

Reviewer's Responses to Questions

**Comments to the Author**

1. Is the manuscript technically sound, and do the data support the conclusions?

Reviewer #1: Yes

Reviewer #2: Partly

2. Has the statistical analysis been performed appropriately and rigorously?

Reviewer #1: Yes

Reviewer #2: Yes

3. Have the authors made all data underlying the findings in their manuscript fully available?

Reviewer #1: Yes

Reviewer #2: Yes

4. Is the manuscript presented in an intelligible fashion and written in standard English?

Reviewer #1: Yes

Reviewer #2: Yes

Reviewer #1: The paper “Prognostic value of FOXA1 in breast cancer estrogen receptor negative: A systematic review and meta-analysis” is a well-written paper. Below are some comments to the authors:

1. In the abstract line 2, if the authors meant the women, then it should be corrected as "especially among those with...".

2. Please, use the full form first and then put acronym in parenthesis in front of the words to use them later, such as Overall-Survival (OS) and Disease-free survival (DFS).

3. In the abstract, it seems like seven studies were used in the Meta Analysis, however the actual results used six. It would be more appropriate to mention the number of studies that was actually used in the Meta Analysis.

4. It seems different treatments were given to the patients in the studies selected and the Mangia study seems to be the only study that included the menopause status in their analysis. It would be interesting to look at the results without Mangia study and compare them with the current results.

5. In section 3.1, it would be easier to follow if a consistent approach is followed in the way of mentioning the numbers in a paragraph, i.e., either use digits or write it out for all.

6. In section 3.4 line 2, it is not clear which four independent studies the authors are referring to. Please, clarify for better readability.

7. There seem to be two versions of tables: Table 1 and Table 2. Please, clarify or correct accordingly.

Reviewer #2: This paper presents a systematic review and meta-analysis investigating the prognostic value of FOXA1 expression specifically in ER-negative breast cancer patients. This is important because the role of FOXA1 on prognosis in hormone receptor-independent environments (ER-negative) had not been extensively explored previously. Current therapies are less effective for ER-negative diagnoses, which often have a worse prognosis, making the identification of new predictive molecules imperative. The study conducted literature searches of multiple databases. Out of 788 initially identified articles, they selected seven studies and reported meta-analysis from six studies. The most important finding is that higher expression of FOXA1 is associated with better overall survival (OS) and disease-free survival (DFS) in estrogen receptor-negative (ER-negative) breast cancer patients.

Major concerns:

Sample size: A significant limitation is the scarcity of available literature on FOXA1 expression in ER-negative breast cancer, with only seven out of 788 articles found for inclusion and the results are derived from six articles (Tables 1 & 2) in the review. One of the seven identified studies had to be excluded from the final analysis because it was not possible to access the complete data required for the review. This further reduced the available evidence.

Heterogeneity: The data reported in these included studies was highly heterogeneous, with different cut-off values used to determine FOXA1 expression. This heterogeneity can complicate the synthesis and interpretation of results. FOXA1 detection methods also contributed to the heterogeneity of the study design: variations in antibody types, concentrations, and potential subjectivity in interpreting IHC results could contribute to the observed heterogeneity. Only one study used PCR for FOXA1 evaluation, there is no comparison of RNA expression can be assessed.

The review found no studies that evaluated the prognostic value of FOXA1 in response to any chemotherapeutic agent in ER-negative patients. This represents a crucial gap, as understanding FOXA1's role in guiding treatment decisions is a key area for any clinical application.

The study explicitly states that research evaluating FOXA1 expression in ER-negative breast cancer cases is "very scarce," this limited pool of studies restricts the breadth and depth of the meta-analysis. The authors acknowledged that the existing evidence is limited and emphasizes the need for additional studies involving larger cohorts to strengthen the validity of the findings. Considering the scarcity of relevant literature, leveraging large-scale omics data (such as genomics, transcriptomics, and proteomics) could offer a more comprehensive assessment of FOXA1's role. Publicly available omics datasets (e.g., TCGA, METABRIC) often include data from thousands of patient samples, allowing for robust statistical analyses and the discovery of associations that might be missed in smaller, heterogeneous clinical studies. This can help overcome the issue of limited sample sizes and data scarcity faced by systematic reviews. It is strongly recommended to include relevant omics data of any type from public domain to provide independent validation of the preliminary literature finding.

**Do you want your identity to be public for this peer review?** For information about this choice, including consent withdrawal, please see our Privacy Policy

Reviewer #1: No

Reviewer #2: No

---

## [Author Response · Author response to Decision Letter 1]

1 Sep 2025

Response to Reviewers

PONE-D-25-39720

Prognostic value of FOXA1 in breast cancer estrogen receptor negative: A systematic review and meta-analysis

PLOS ONE

Dear Dr. Yan Xu, Ph.D.,

Academic Editor,

PLOS ONE

Editor Comment: As reviewers pointed out, while systematic reviews are crucial for synthesizing existing clinical evidence, their limitations, particularly concerning data scarcity and heterogeneity, can be significant. Integrating findings from systematic reviews with insights derived from large-scale omics data analysis could provide a more complete and nuanced understanding of biomarkers like FOXA1 in complex diseases such as breast cancer.

Response: We appreciate your valuable suggestion. We agree with its potential to enrich our understanding of biomarkers. However, our study design strictly adheres to a predefined protocol, registered and developed according to the Cochrane for Systematic Reviews (https://www.cochrane.org/authors/handbooks-and-manuals/handbook/current), the JBI Evidence Synthesis Manual Home (https://jbi-global-wiki.refined.site/space/MDJPLSDLE/217940062/Manual+del+JBI+para+la+S%C3%ADntesis+de+la+Evidencia+Home), and PRISMA methodological guidelines for systematic reviews and meta-analyses. As you are aware, these guidelines standardize the synthesis of existing clinical evidence; however, they do not encompass the generation or integration of primary omics data, which would necessitate a distinct experimental and analytical design. Therefore, to maintain the integrity and methodological rigor of our work, we focused on synthesizing clinical evidence. However, following your suggestion, we include in our recommendations the conduct of omics studies as the next logical and necessary step arising from our conclusions in the General Recommendation Section:

General Recommendation

Studies based on omics data are needed that do not limit current publications and allow for a more complete understanding of relevant biomarkers, such as FOXA1, in complex diseases such as estrogen receptor-negative breast cancer. Furthermore, further studies in larger cohorts are needed to understand the relationship between FOXA1 expression and breast cancer prognosis in hormone-independent settings, as well as its association with treatment response, using standardized assessment techniques.

Editor Comment: Authors used many acronyms in the text, which makes it hard for the general readers. An acronyms index or table will be useful.

Response: To make the text readable and accessible to all audiences, we decided to remove the acronyms and use the full text instead. The corrections were underlined in the document.

Editor Comment: Please ensure that your manuscript meets PLOS ONE's style requirements, including those for file naming. The PLOS ONE style templates can be found

Response: The manuscript has been fully revised and adjusted in accordance with PLOS ONE guidelines.

Editor Comment: Please state what role the funders took in the study.

Response: We clarify that this work was supported by the Foundation for the Promotion of Research and Technology - Banco de la República (Grant No. 5,038) and Ministerio de Ciencia, Tecnología e Innovación de Colombia for the National Doctorate scholarship for Teachers 909-2022. The funders had no role in study design, data collection, analysis, the decision to publish, or preparation of the manuscript. This funding supported educational scholarships for one author and was not directly related to this research project. For this reason, we noted in section A 7. Acknowledgments:

Section: Acknowledgements: Thanks to the Ministerio de Ciencia, Tecnología e Innovación de Colombia for the National Doctorate scholarship for Teachers 909-2022 by Angela V. Fonseca-Benítez. Foundation for the Promotion of Research and Technology - Banco de la República (Grant No. 5,038).

Editor Comment: Please note that funding information should not appear in any section or other areas of your manuscript. We will only publish funding information present in the Funding Statement section of the online submission form. Please remove any funding-related text from the manuscript.

Response: Financing information was removed from the document and added to Acknowledgments.

Editor Comment: Please include a separate caption for each figure in your manuscript.

Response: In response to the Editor's comment, we have now provided separate captions for Figures 3 and 4.

Editor Comment: Please include your tables as part of your main manuscript and remove the individual files. Please note that supplementary tables (should remain/ be uploaded) as separate "supporting information" files.

Section: Supporting Information

S1 Table: Database Search Term

S2 Table: List of excluded items and reason for exclusion

Figure S1: Sensitivity analysis. Forest plot of Disease-free survival as a function of FOXA1 expression in estrogen receptor-negative breast cancer CI: Confidence interval. Excluded Mangia.

Response: Thank you for this clarification; we have now integrated all main tables directly into the manuscript file and have ensured that any supplementary tables remain uploaded separately as supporting information files.

Editor Comment: Please remove all personal information, ensure that the data shared are in accordance with participant consent, and re-upload a fully anonymized data set.

Response: Thank you very much for your thoughtful comment and for providing detailed guidance on data preparation. We sincerely appreciate the opportunity to clarify that our manuscript is a systematic review. As such, it does not report on a primary study involving human participants, and therefore, no original datasets were generated or analyzed. All the data presented were thoughtfully extracted from previously published and publicly available studies, which are comprehensively cited in our reference list. Consequently, the concerns regarding participant consent and personal information do not apply to our work. We are grateful for your vigilance on this important matter and hope this clarification is helpful.

Editor Comment: Please include captions for your Supporting Information files at the end of your manuscript, and update any in-text citations to match accordingly. Please see our Supporting Information guidelines for more information: http://journals.plos.org/plosone/s/supporting-information.

Response: Thanks a lot for your comment. The supporting information has been updated in accordance with the journal's guidelines

Reviewers' comments:

Reviewer #1:

We appreciate your comments, which have helped us improve the accuracy of our manuscript. We will now address them:

Comment 1: In the abstract line 2, if the authors meant the women, then it should be corrected as "especially among those with...".

Response: The paragraph was corrected to: Breast cancer continues to be the primary type of cancer in the world in women, and current therapies still increase the rates of recurrence of the disease, especially among those with negative estrogen receptor diagnoses, where current treatment is less effective and has been shown to have a worse prognosis.

Comment 2: Please, use the full form first and then put acronym in parenthesis in front of the words to use them later, such as Overall-Survival (OS) and Disease-free survival (DFS).

Response: Thank you for your feedback. To improve readability, we have spelled out all acronyms upon their first use and underlined these changes for your convenience. As shown in the example below, we use disease-free survival or overall survival in all instances throughout the manuscript.

Comment 3: In the abstract, it seems like seven studies were used in the Meta Analysis, however the actual results used six. It would be more appropriate to mention the number of studies that was actually used in the meta-analysis.

Response: Thank you for your attention to detail. You are correct; the meta-analysis included seven studies. The seventh study was inadvertently omitted from the initial table during formatting. We have now updated the document to include the complete data set. The studies used for OS were Albergaria, 2009; Dai, 2019; Chen, 2021, while the studies used for DFS Chen, 2021; Dai, 2019; Hisamatsu, 2015; Mangia, 2019; Jing, 2019; Zhou, 2023. (Table 1 and 2). This correction was made in the results section.

Comment 4: It seems different treatments were given to the patients in the studies selected and the Mangia study seems to be the only study that included the menopause status in their analysis. It would be interesting to look at the results without Mangia study and compare them with the current results.

Response: We thank the reviewer for this excellent suggestion. We have performed a sensitivity analysis by excluding the study by Mangia et al. The results of this sensitivity analysis showed no substantial change in the overall effect estimate (0.65), confirming the robustness of our primary findings. The new results have been added to the manuscript as Supporting Figure 1. The description of these results was added to the manuscript in the results section and the figure into the supporting information. For better revision, we copied the figure below.

Comment 5: In section 3.1, it would be easier to follow if a consistent approach is followed in the way of mentioning the numbers in a paragraph, i.e., either use digits or write it out for all.

Response: Thank you for this feedback. We have revised Section 3.1 to ensure all numerical values are now presented consistently in numeral form throughout the paragraph for improved clarity and flow.

Section Methods: Search and study selection process: Using the search strategy described above, a total of 788 potentially eligible articles were identified from the various databases used: Embase (402), PubMed (102), Scopus (106), BSV (130), Google Scholar (81), Lilacs (0), and Cochrane (20). 161 articles were removed during the duplicate review process. 627 documents were screened by title and abstract for the evaluation process, where 531 were excluded for failing to meet the eligibility criteria. Finally, 96 articles were evaluated for definitive review in full text, of which 8 studies were included [11-18] (Fig 1). However, one of these was excluded since it was not possible to access the data required for the review. The studies excluded during the final selection phase are presented in Sup File 2.

Comment 6: In section 3.4 line 2, it is not clear which four independent studies the authors are referring to. Please, clarify for better readability.

Response: We thank the reviewer for this observation. We have clarified the sentence by explicitly naming the three studies: The studies used for Overall Survival were Albergaria, 2009; Dai, 2019; Chen, 2021, while the studies used for Disease-Free Survival were Chen, 2021; Dai, 2019; Hisamatsu, 2015; Mangia, 2019; Jing, 2019; Zhou, 2023. This correction was made in Results Section.

Comment 7: There seem to be two versions of tables: Table 1 and Table 2. Please clarify or correct accordingly.

Response: Thank you for pointing that out. Indeed, there was an error during formatting which caused the tables to be truncated. This has now been fixed in the document.

Reviewer #2:

Comment 1 - Sample size: A significant limitation is the scarcity of available literature on FOXA1 expression in ER-negative breast cancer, with only seven out of 788 articles found for inclusion, and the results are derived from six articles (Tables 1 & 2) in the review. One of the seven identified studies had to be excluded from the final analysis because it was not possible to access the complete data required for the review. This further reduced the available evidence.

Response: We sincerely thank the Reviewer for their insightful comments, which have provided us with the opportunity to clarify this important aspect of our methodology. Our systematic literature search was designed to be comprehensive across all breast cancer subtypes. However, the eligibility criteria required that studies specifically focus on estrogen receptor (ER)-negative disease and report data on the primary endpoints of overall survival (OS) and disease-free survival (DFS). The application of these stringent criteria identified seven eligible articles for inclusion. We acknowledge that the final number of studies is limited. This was a deliberate consequence of our strategy to ensure a homogeneous patient population and the consistent reporting of outcomes relevant to our research question. We have transparently addressed the potential limitations arising from this approach, including the limited number of studies, in the section on Limitations. As stated there, we believe these findings highlight the need for more extensive research in this specific population to build upon this work and reach more definitive conclusions.

Limitations

Systematic reviews are inherently limited by the quality of the primary studies included. A significant limitation is the high heterogeneity among current studies, particularly in the techniques and protocols used for FOXA1 determination (e.g., methodologies, antibodies, and cutoff values). Furthermore, there are pronounced differences in the statistical analyses performed in the included studies, complicating comparability and the pooled interpretation of results. Another critical limitation is that the impact of FOXA1 expression on posttreatment prognosis in hormone-independent settings is not yet fully elucidated. Therefore, we advocate for future standardized studies to address these gaps. This research should place special emphasis on assessing chemotherapy response in various breast cancer subtypes, particularly aggressive phenotypes such as basal-like or triple-negative cancers, which are associated with lower survival rates. Ultimately, additional studies with larger, well-defined cohorts are essential to reinforce the validity of these findings. Therefore, the conclusions of this study must be considered within the context of its limitations. The generalizability of the findings is limited by heterogeneity in FOXA1 assay methodologies and scoring criteria, and these results should therefore be interpreted with caution.

Comment 2 - Heterogeneity: The data reported in these included studies were highly heterogeneous, with different cut-off values used to determine FOXA1 expression. This heterogeneity can complicate the synthesis and interpretation of results. FOXA1 detection methods also contributed to the heterogeneity of the study design: variations in antibody types, concentrations, and potential subjectivity in interpreting IHC results could contribute to the observed heterogeneity. Only one study used PCR for FOXA1 evaluation; there is no comparison of RNA expression can be assessed.

Response: We appreciate the reviewer for this critical observation. We fully acknowledge that methodological heterogeneity in FOXA1 detection is a major contributor to the overall study heterogeneity. As rightly noted, variations in IHC protocols (antibodies, clones, concentrations, scoring systems) introduce significant inter-study variability. However, in our methodology, we included articles that identified FOXA1 by both PCR and IHC, since this could also affect the number of articles obtained for the final analysis. We understand that the fact that only one study utilized a PCR-based method indeed precludes any robust comparative analysis between protein (IHC) and RNA expression levels, which is a considerable limitation. In the revised discussion, we explicitly state this as a key source of heterogeneity and a limitation of our study. Furthermore, we emphasize the pressing need for standardized, reproducible assays (like RNA-seq or quantitative PCR) in future studies

---

## [Decision Letter · Decision Letter 1]

18 Sep 2025

Dear Dr. Rangel,

Thank you for submitting your manuscript to PLOS ONE. After careful consideration, we feel that it has merit but does not fully meet PLOS ONE’s publication criteria as it currently stands. Therefore, we invite you to submit a revised version of the manuscript that addresses the points raised during the review process.

We look forward to receiving your revised manuscript.

Kind regards,

Yan Xu, Ph.D.

Academic Editor

PLOS ONE

Journal Requirements:

Reviewers' comments:

Reviewer's Responses to Questions

**Comments to the Author**

Reviewer #3: (No Response)

2. Is the manuscript technically sound, and do the data support the conclusions?

Reviewer #3: Yes

3. Has the statistical analysis been performed appropriately and rigorously?

Reviewer #3: Yes

4. Have the authors made all data underlying the findings in their manuscript fully available?

Reviewer #3: Yes

5. Is the manuscript presented in an intelligible fashion and written in standard English?

Reviewer #3: No

Reviewer #3: In the abstract please revise the sentence stating treatment increases the risk of recurrence.

Figure 1: fix the n= formatting in the excluded studies box for consistency on each line.

In the paragraph about the risks of studies, say 6 out of 7 rather (or as well as) than 88%. And the same for the other percentages.

Table 2: have the appropriate use of . For decimals for this audience (not ,)

Below figure 3 hazard rather than hazzart.

**Do you want your identity to be public for this peer review?** For information about this choice, including consent withdrawal, please see our Privacy Policy

Reviewer #3: No

---

## [Author Response · Author response to Decision Letter 2]

19 Sep 2025

Response to Reviewer

PONE-D-25-39720

Prognostic value of FOXA1 in breast cancer estrogen receptor negative: A systematic review and meta-analysis

PLOS ONE

Dear Dr. Yan Xu, Ph.D.,

Academic Editor,

PLOS ONE

We thank the reviewer for their thorough and constructive feedback. We appreciate the time and expertise dedicated to reviewing our manuscript. All comments have been addressed point-by-point below, and we believe the suggested revisions have significantly improved the quality and clarity of our paper.

Is the manuscript presented in an intelligible fashion and written in standard English? Reviewer #3: No

Response: The manuscript was reviewed, and paragraph-by-paragraph corrections were made using the Grammarly Pro tool. During this process, the title was modified, and the relevant changes were highlighted.

Old title: Prognostic value of FOXA1 in breast cancer estrogen receptor negative: A systematic review and meta-analysis

New title: Prognostic Value of FOXA1 in Estrogen Receptor-Negative Breast Cancer: A Systematic Review and Meta-Analysis.

Comment Reviewer #3: In the abstract please revise the sentence stating treatment increases the risk of recurrence.

Response: In the abstract, the phrase "treatment increases the risk of recurrence" was changed to a more precise and scientifically correct phrase: "recurrence rates remain high despite current therapies for the disease."

Comment Reviewer #3 Figure 1: fix the n= formatting in the excluded studies box for consistency on each line.

Response: We appreciate your attention to detail, the format of n= in Figure 1: has been corrected.

Comment Reviewer #3: In the paragraph about the risks of studies, say 6 out of 7 rather (or as well as) than 88%. And the same for the other percentages.

Response: As requested by the reviewer, the risk percentages in the indicated paragraph the risks of studies have been revised to reflect the actual number of articles.

Comment Reviewer #3 Table 2: have the appropriate use of . For decimals for this audience (not ,)

Response: We thank the reviewer for their attention to detail. The respective corrections have been made in Table 2, and the changes have been highlighted for ease of review.

Comment Reviewer #3: Below figure 3 hazard rather than hazzard.

Response: We thank the reviewer for pointing out this typographical error. The misspelling of 'hazard' has been corrected in the text below Figure 3.

---

## [Editor Report · Decision Letter 2]

30 Sep 2025

Prognostic Value of FOXA1 in Estrogen Receptor-Negative Breast Cancer: A Systematic Review and Meta-Analysis.

PONE-D-25-39720R2

Dear Dr. Rangel,

We’re pleased to inform you that your manuscript has been judged scientifically suitable for publication and will be formally accepted for publication once it meets all outstanding technical requirements.

Kind regards,

Yan Xu, Ph.D.

Academic Editor

PLOS ONE
---

## [Editor Report · Acceptance letter]

PONE-D-25-39720R2

PLOS ONE

Dear Dr. Rangel,

I'm pleased to inform you that your manuscript has been deemed suitable for publication in PLOS ONE. Congratulations! Your manuscript is now being handed over to our production team.

Kind regards,

on behalf of

Dr. Yan Xu

Academic Editor

PLOS ONE